# Relationship of Psychological Flexibility and Mindfulness to Caregiver Burden, and Depressive and Anxiety Symptoms in Caregivers of People with Dementia

**DOI:** 10.3390/ijerph20054232

**Published:** 2023-02-27

**Authors:** Khai Pin Tan, Jin Kiat Ang, Eugene Boon Yau Koh, Nicholas Tze Ping Pang, Zanariah Mat Saher

**Affiliations:** 1Department of Psychiatry and Mental Health, Hospital Tengku Ampuan Afzan, Kuantan 25100, Pahang, Malaysia; 2Department of Psychiatry, Universiti Putra Malaysia, Serdang 43400, Selangor, Malaysia; 3Faculty of Medicine and Health Sciences, Universiti Malaysia Sabah, Kota Kinabalu 88400, Sabah, Malaysia; 4Department of Psychiatry and Mental Health, Kuala Lumpur General Hospital, Kuala Lumpur 50586, Federal Territory of Kuala Lumpur, Malaysia

**Keywords:** dementia, psychological flexibility, mindfulness, caregiver burden, depression, anxiety

## Abstract

Caregivers of People with dementia (PwD) commonly experience burdens and other mental health issues, e.g., depression and anxiety. At present, there are limited studies that examine the relationships between caregiver psychological factors and caregiver burden, and depressive and anxiety symptoms. Therefore, this study’s objectives were to examine the relationships between psychological flexibility and mindfulness in caregivers of PwD, and to determine the predictors of these three outcomes. This was a cross-sectional study conducted in the geriatric psychiatry clinic of Kuala Lumpur Hospital, Malaysia, and the sample (*n* = 82) was recruited via a universal sampling method over three months. The participants completed a questionnaire that consisted of the sociodemographics of the PwD and caregivers, illness characteristics of the PwD, Acceptance and Action Questionnaire-II (AAQ-II), Mindful Attention Awareness Scale (MAAS), Zarit Burden Interview Scale (ZBI), Patient Health Questionnaire-9 (PHQ-9) and Generalized Anxiety Disorder-7 (GAD-7). The results show that despite significant relationships between psychological flexibility and mindfulness and lower levels of caregiver burden, and depressive and anxiety symptoms (*p* < 0.01), only psychological inflexibility (*p* < 0.01) remained as a significant predictor of the three outcomes. Therefore, in conclusion, intervention programs that target the awareness of the caregiver’s psychological inflexibility should be implemented to alleviate these adverse outcomes in dementia caregivers.

## 1. Introduction

There are 55 million people worldwide who have dementia, equivalent to 5–8% of the elderly population (60 years old and above) worldwide [1]. The number of PwD is expected to rise to 78 million by 2030 and 139 million by 2050 due to the global population aging phenomenon. The drastic increment in life expectancy is generally a welcome trend, but the challenge of the increasing prevalence of dementia is alarming. According to Alzheimer’s disease International, the estimated number of PwD in Malaysia was 123,000 in 2015, and it is expected to reach 261,000 by 2030 and 590,000 by 2050 [2]. Malaysia’s overall prevalence of probable dementia is about 8.5% [3]. It is one of the most significant health and social care challenges in this 21st century globally because it is a significant cause of disability and dependency in the elderly, contributing to 11.9% of the overall disability due to non-communicable diseases [4]. Our local study on the Malaysian burden of disease and injury was in keeping with the worldwide finding; in 2017, it was estimated that dementia was the second and third leading cause of disability burden among males and female elderly above 80 years, respectively [5].

Being an illness that significantly interferes with independence in everyday activities, most PwD require assistance and care from others. Therefore, dementia has a substantial impact on caregivers, families, communities, societies, and governments, especially in terms of increased costs and decreased productivity for economies. It is estimated that each year, the global cost of dementia is about USD 818 billion, where 85% of the total cost is related to the caregiver, family, and social services rather than medical care [6,7]. Without a doubt, the involvement of caregivers improves the quality of life (QOL) of PwD, delays the need for institutional care for PwD, and reduces the governmental financial burden [8]. However, they are also likely to experience more significant perceived burden and depressive symptoms compared with caregivers of people without dementia due to various factors, e.g., greater variety of care tasks, greater employment complications, and longer durations of care, i.e., more than 40 h care per week, and greater financial burden [9,10,11]. Nearly a quarter of caregivers of PwD provide 40 h of care or more per week, compared with 16% for non-dementia caregivers [8].

Caregiver burden is a common phenomenon in Malaysia, an Asian country where the filial obligation is a cultural norm. Family members are generally socially assigned, morally obliged, and inherently assumed to be responsible for taking care of their unwell family members [12,13]. A significant number of elderly live together with their children, especially after their spouses have passed on, but with the massive urban migration (both interurban migration and rural–urban migration) among the younger adults for better occupational opportunities, they have been experiencing increased challenges in caring for the older generations in their families [14,15,16]. Nevertheless, the inflation and higher living costs in the cities, coupled with slow economic growth, has caused urban poverty, where financial constraints further increase the burden on caregiving. Regardless of their actual burden level, the caregivers do not share it or seek help voluntarily due to the fear of family disputes or being labeled as ungrateful children, and their resentment over caregiving is secondary [12].

Caregivers of PwD experience higher burden levels than caregivers of people with other diseases. Globally, dementia has significantly contributed to a significant burden in almost 50% of caregivers, and what is more concerning and alarming is that the burden level tends to escalate over time [17]. It is believed that there are more caregivers suffering from caregiver burden than the numbers reported, due to the increased prevalence of dementia. A local study reported a mean score of 35.4 on the Zarit Burden Interview Scale (ZBI), indicating that caregivers of PwD experience a moderate amount of burden, and 5.7% were severely burdened [18]. Unfortunately, no updated local prevalence is available.

The caregiver burden is not the only mental health outcome that is alarming and worrying. A significant number of caregivers also suffer from depression and anxiety. The prevalence of depressive and anxiety symptoms in dementia caregivers ranges from 29.9% to 74% and 7% to 44%, respectively, which is much higher than in the general population [19,20,21,22,23,24,25]. The presence of depressive or anxiety symptoms does not indicate a person suffers from depressive or anxiety disorders if the symptoms do not cause significant distress or impairment in functioning [26]. Except for a few studies, most of the literature explores and reports the outcomes as depressive and anxiety symptoms. The literature reports an incidence rate of 37% for major depressive disorder after a 24-month interval of caregiving, 55% for anxiety disorder, and 60% for comorbid depressive and anxiety disorders [27]. Dementia caregivers are also at five times higher risk of fulfilling the diagnosis of major depressive disorder than the general population [28]. Caregivers of PwD experience a more significant impact on their physical and mental health, with lesser time to access support and a higher risk of morbidity and mortality.

Most of the studies explore the relationships between the dementia patient and caregiver sociodemographics, patient illness characteristics, and caregiver burden and depressive and anxiety symptoms. Illness severity and presence of behavioural and psychological symptoms of dementia (BPSD) are known predictors for caregiver burden and depressive and anxiety symptoms [25,29,30,31,32,33,34,35,36,37,38,39,40,41,42,43,44,45,46,47,48]. PwD and the caregivers’ sociodemographic background have also been studied in Malaysia and in other countries [18,44,49]. Lower patient and caregiver educational attainment (i.e., explained by lower brain plasticity and cognitive reserves), lower caregiver socioeconomic and financial status, longer duration of caregiving, and unshared caregiving or poorer social support are some of the established risk factors for caregiver burden, and depressive and anxiety symptoms [25,30,33,35,36,41,44,50,51,52,53,54,55,56,57,58]. Other sociodemographic factors were inconsistent.

However, many of these characteristics were non-modifiable or indirect factors, which are relatively more challenging for intervention. More studies explored various psychological factors of the caregivers in relationship with their mental health outcomes. Among all, coping strategies are more commonly studied. It is well established that problem-focused coping strategies, social-emotional support coping strategies, and acceptance are significantly associated with a lower level of burden, anxiety, and depressive symptoms among dementia caregivers [59,60,61,62]. On the other hand, dysfunctional coping strategies, avoidance, and wishful thinking are associated with worse mental health outcomes. Therefore, various psychological interventions have been recommended, for example, psychoeducation, behavioural therapy, support groups, individual and family counselling, and multicomponent interventions [63,64,65,66]. For example, a local systematic review reported that despite using different methods, both group and individual education interventions helped reduce the psychological distress of the informal dementia caregiver [67]. However, the treatment size in these studies was relatively small, and the results were inconsistent [68,69,70,71]. Other psychological factors, e.g., psychological flexibility and mindfulness, the core components in the trending third-wave psychotherapies, have not been widely explored, especially in regional and local settings.

Even though it might appear that mindfulness and psychological flexibility are interrelated, both constructs explained unique interindividual variations in psychological health, and they reflect a significant amount of information that is not overlapping [72]. Psychological flexibility is the ability to act consistently with long-term goals and values, despite inner discomfort or unwanted events [73]. It is also related to experiential acceptance. Mindfulness is otherwise referred to as bringing attention to present experiences intentionally without being judgemental [74,75]. Psychological flexibility has been studied for its relationship with various mental health outcomes among patients, e.g., depression, anxiety, psychosis, epilepsy, pain, etc., and caregivers of patients with psychosis and Parkinson’s disease [76,77]. However, limited literature is available on its relationship with dementia caregiver burden, and depressive and anxiety symptoms [78,79,80]. Studies exploring the relationship between mindfulness and caregiver mental health are even scarcer [81,82]. Even though the findings from previous studies were significant in general, there are no regional or local studies available. Therefore, mindfulness is believed to be related to a lower level of burden, and reduced depressive and anxiety symptoms in dementia caregivers. It could, therefore be applied in interventions to reduce distress and improve the well-being of caregivers.

To the researcher’s knowledge, this study is the first attempted in Malaysia or in Asia more widely that studies the relationship between psychological flexibility and mindfulness and the level of burden, and depressive and anxiety symptoms in caregivers of PwD. Limited studies have been conducted among the caregivers of PwD. The results are expected to contribute to future recommendations for providing feasible and effective psychological interventions that target the particular psychological factors that impact caregivers. The findings from this study can be used to suggest routine screening and early detection of caregiver burden, and depressive and anxiety symptoms, and potentially related psychological factors, hence reducing their burden level and improving their QOL and well-being in caring for the PwD.

## 2. Materials and Methods

This study was a hospital-based, cross-sectional observational study, conducted in the geriatric psychiatry clinic, Kuala Lumpur General Hospital, Malaysia. The sample size was calculated using a meta-analysis approach because Cochran’s sample size formula was not appropriate (the prevalence of caregiver burden could not be determined due to its subjectiveness, and independent variables were not considered). The required sample size was eventually obtained from the Cohen Power Table, where the Pearson’s *r* was 0.27. With a power of 0.80, the estimated required sample size was 85. Eventually, a total of 82 caregivers of PwD attending the geriatric psychiatry clinic were recruited via a universal sampling method based on the records of the geriatric psychiatry clinic over three months from October 2021 to January 2022. The inclusion criteria were caregivers of PwD aged 18 years and above who provided care for at least three months [83] and met at least 2 (for non-relatives) or 3 (for relatives) of the following criteria established by Pollak and Perlick [84]: (1) the caregiver is a spouse, parent, or spouse equivalent; (2) has the most frequent contact with the patient; (3) helps to support the patient financially; (4) has most frequently been collateral in the patient’s treatment; and (5) is a contact for treatment staff in case of emergency. The final inclusion criterion was that they were capable to give written informed consent. Apart from caregivers who did not fulfil the inclusion criteria, caregivers with PwD living in nursing homes, caregivers of PwD whom had passed away, and formal caregivers (defined as a person hired and paid to care for a dementia patient in exchange for remuneration, i.e., registered nurses, domestic workers, social workers, etc.) were excluded from this study. The eligible participants provided informed written consent and completed a pre-tested, validated, self-administrated online questionnaire collecting the patients’ and caregivers’ sociodemographic information, the patients’ illness characteristics, AAQ-2, MAAS, ZBI, PHQ-9 and GAD-7.

The sociodemographic information collected from PwD included age, gender, ethnicity, religion, marital status and education level. The collected sociodemographics for caregivers included age, gender, ethnicity, religion, marital status, education level, employment status, monthly household income, perceived financial status, caregiver’s relationship with PwD, caregiving duration, and sharing of caregiving. In term of illness characteristics of PwD, illness duration, history of psychiatric ward admission, types of dementia, illness severity and BPSD were measured in the study.

AAQ-II is one of the most widely used questionnaires to assess the construct referred to as acceptance, psychological inflexibility, or experiential avoidance [85]. It is a 7-item Likert-type self-rated questionnaire. It contains items on negative evaluations of feelings, avoidance of thoughts and feelings, and differentiates a thought from its referent and behavioural adjustment in the presence of difficult internal experiences, such as thoughts, sensations, and feelings. Higher total scores indicate higher psychological inflexibility and experiential avoidance. AAQ-II has shown good internal consistency (α = 0.84 (0.78–0.88)) and good test-retest reliability over 3 and 12 months (0.81 and 0.79 respectively) [85]. The Cronbach alpha of AAQ-II in the present study was 0.94, with all items ranging between 0.93 and 0.94.

MAAS is a 15-item Likert-type self-rated questionnaire most frequently used in measuring mindfulness. It is designed to assess the individual’s attention and awareness in everyday life, as it focuses on the attention awareness component of the mindfulness construct and measures the frequency of the mindful states of an individual in his or her day-to-day life [86]. In contrast to AAQ-II, higher total scores indicate a greater mindfulness level for MAAS. It has demonstrated good psychometric properties with high validity (α ≥ 0.82) and reliability (i.e., 4-week test-retest reliability (interclass *r* = 0.81)) [87]. In the present study, the Cronbach alpha of MAAS was 0.95, with all items ranging between 0.94 and 0.95.

Being the commonest questionnaires used globally to measure caregiver burden, ZBI is a Likert-type self-rated questionnaire consisting of 22 items. The higher the score, the greater the burden a caregiver is suffering from. It is a validated and reliable instrument with a Cronbach’s alpha value of 0.92–0.93 and an intra-class coefficient for test-retest reliability of 0.89 [88]. In the present study the Cronbach alpha of ZBI was 0.94, with all items ranging between 0.93 and 0.94.

PHQ-9 is the most widely used questionnaire to measure depressive symptoms [89]. It incorporates the DSM-IV depression criteria with other leading major depressive symptoms. It is a Likert-type self-rated 9-item questionnaire. The higher the total score, the more severe the depressive symptoms an individual is experiencing. It is highly valid and reliable, with good internal consistency (α = 0.87) and test-retest reliability of 0.73 [90,91]. The Cronbach alpha value for PHQ-9 in this study was 0.88, where all items ranged between 0.86 and 0.88.

GAD-7, a 7-item self-rated questionnaire, was also used in this study to measure anxiety symptoms [92]. It reflects the DSM-IV criteria for generalised anxiety disorder. Higher total scores reflect higher anxiety symptoms. It is a valid and reliable instrument with good internal consistency (α = 0.92) and convergent and construct validity [92,93,94,95]. The Cronbach alpha value for GAD-7 in this study was 0.92, and each item was between 0.92 and 0.93.

The data were analyzed using the Statistical Package for the Social Sciences (SPSS) version 25.0. All the categorical independent variables were statistically analyzed with parametric tests, either independent *t*-test or one-way ANOVA. All the continuous independent variables, except caregiver monthly household income, were analyzed with Pearson correlation test due to their normally distributed properties. Caregiver monthly household income, the one and only non-normally distributed independent variable, was analyzed with Spearman correlation test. A multicollinearity test prevented highly correlated factors being included in the model before the multivariable analysis. A variance inflation factor of <5 indicated there was no multicollinearity issue. Hierarchical general linear model analysis was then used to determine the significant predictors of caregiver burden in caregivers of PwD. All *p*-values less than 0.05 level were considered as significant.

## 3. Results

### 3.1. Descriptive Statistics

#### 3.1.1. Sociodemographics of PwD

In regard to the sociodemographics of PwD, the mean age of PwD was 77.84 years (SD = 6.70), and female patients were slightly more numerous than male patients (57.3% vs. 42.7%). The rest of the sociodemographic information are tabulated in Table 1.

#### 3.1.2. Sociodemographics of Caregivers of PwD

Table 2 displays the sociodemographic information of the caregivers of PwD. The average age of dementia caregivers was 51.5 years (SD = 14.63), and more than two-thirds of the caregivers (68.3%) were females. Their median monthly household income was 725.85 USD (IQR = 752.74 USD). In addition, only 31.7% of the caregivers perceived that they had a problematic financial status. Regarding their caregiving role for PwD, more than half were children (59.8%), and a quarter were spouses or partners (25.6%). They provided care for the PwD 14.65 h a day on average (SD = 8.39), and 72.0% had someone else to share the caregiving role.

#### 3.1.3. Illness Characteristics of PwD

The mean duration of dementia among the PwD was 5.88 years (SD = 3.64), and only 7.3% of them had been admitted once or more to the psychiatric ward during their course of illness. The top three commonest aetiologies of dementia among the study population were Alzheimer’s disease (47.6%), Mixed Vascular-Alzheimer Dementia (MVAD) (30.5%), and vascular dementia (17.1%). The severity of dementia was heterogeneous: 37.8% were mild, 30.5% were moderate, and 31.7% were severe. Among the PwD, 70.7% had BPSD. The results are shown in Table 3.

#### 3.1.4. Psychological Flexibility and Mindfulness of Caregivers of PwD

Table 4 shows the level of psychological flexibility and mindfulness of caregivers of PwD. The mean score for AAQ-II, which measures psychological flexibility, was 22.45, and the mean score for MAAS measuring mindfulness was 68.23.

#### 3.1.5. Caregiver Burden, and Depressive and Anxiety Symptoms in Caregivers of PwD

The caregivers of PwD were screened for any significant caregiver burden, and depressive and anxiety symptoms. The recommended cut-off points for assessment scales were used to determine if the symptoms were significant. Otherwise, the data was retained as continuous data for later statistical analysis. The mean score for ZBI, which measures caregiver burden level, was 33.17, and with the cut-off point of 22, 67.1% [96] of the caregivers reported significant burden. The mean score for PHQ-9, which measures depressive symptoms, was 7.10, and 30.5% of the dementia caregivers experienced significant depressive symptoms, based on the cut-off point of 10 [97]. The mean score on GAD-7, which measures anxiety symptoms, was 5.41. Based on a cut-off point of 10, 20.7% reported significant anxiety symptoms [91]. The results are displayed in Table 5.

### 3.2. Bivariate Analysis of Factors Related to Caregiver Burden, and Depressive and Anxiety Symptoms in Caregivers of PwD 

Bivariate analyses were performed to investigate the relationships between sociodemographic factors of the PwD and the caregivers, the illness characteristics of the PwD, and the psychological factors of caregivers, and the level of caregiver burden, and depressive and anxiety symptoms.

#### 3.2.1. Relationships between Patient Sociodemographics and Caregiver Burden, and Depressive and Anxiety Symptoms in Caregivers of PwD

The relationships between patient sociodemographics and caregiver burden, and depressive and anxiety symptoms in caregivers of PwD are displayed in Table 6. None of the patient sociodemographics were related to caregiver burden and anxiety symptoms in caregivers of PwD. However, the patient’s age had a significant relationship with depressive symptoms in dementia caregivers, where caregivers of younger PwD suffered from a significantly higher level of depressive symptoms (*r* = −0.22, *p* = 0.04). The results are shown in Table 6.

#### 3.2.2. Relationships between Caregiver Sociodemographics and Caregiver Burden, and Depressive and Anxiety Symptoms in Caregivers of PwD

There were no significant relationships between the various caregiver sociodemographics and caregiver burden. Caregiver’s age, caregiver’s employment status, and the caregiver–patient relationship were significantly related to depressive and anxiety symptoms in caregivers of PwD. Firstly, younger dementia caregivers suffered from significantly greater depressive (*r* = −0.36, *p* = 0.00) and anxiety symptoms (*r* = −0.27, *p* = 0.02). There were significant relationships between caregiver’s employment status and depressive (*p* = 0.01) and anxiety symptoms (*p* = 0.02) in caregivers of PwD. Compared with retired caregivers, caregivers who were full-time employed suffered from significantly greater depressive symptoms (*p* = 0.02), whereas unemployed caregivers suffered from significantly greater anxiety symptoms (*p* = 0.04). Lastly, non-spousal caregivers were also identified to be experiencing significantly higher levels of depressive (*p* = 0.00) and anxiety symptoms (*p* = 0.04) than spousal caregivers. The relationships between caregiver sociodemographics and burden, and depressive and anxiety symptoms are tabulated in Table 7.

#### 3.2.3. Relationships between Patient Illness Characteristics and Caregiver Burden, and Depressive and Anxiety Symptoms in Caregivers of PwD

Table 8 shows the relationships between patient illness characteristics and caregiver burden, and depressive and anxiety symptoms in caregivers of PwD. There was no significant relationship between patient illness characteristics and caregiver burden and depressive symptoms in caregivers. However, disease severity was significantly related to caregiver anxiety symptoms, where caregivers of patients with mild dementia suffered from higher levels of anxiety symptoms than caregivers of patients with severe dementia (*p* = 0.02).

#### 3.2.4. Relationships between Psychological Flexibility and Mindfulness with Caregiver Burden, and Depressive and Anxiety Symptoms in Caregivers of PwD

The relationships between psychological flexibility and mindfulness with caregiver burden, and depressive and anxiety symptoms in caregiver of PwD are shown in Table 9. There were significant relationships between psychological flexibility and burden (*r* = 0.74, *p* = 0.00), depressive symptoms (*r* = 0.61, *p* = 0.00), and anxiety symptoms (*r* = 0.69, *p* = 0.00) in caregivers of PwD. Psychological inflexibility was related to higher level of caregiver burden, depressive symptoms and anxiety symptoms. Mindfulness was also found to be significantly related to lower levels of burden (*r* = −0.37, *p* = 0.00), depressive symptoms (*r* = −0.43, *p* = 0.00), and anxiety symptoms (*r* = −0.43, *p* = 0.00) in caregivers of PwD.

### 3.3. Multivariate Analysis on Predictors of Caregiver Burden, and Depressive and Anxiety Symptoms in Caregivers of PwD

The data were further analyzed with multivariate analysis to identify the predictors for the level of burden, and depressive and anxiety symptoms among the caregivers of PwD. Few variables from the bivariate analyses were found to have significant relationships to the outcomes with a *p*-value < 0.05. Therefore, variables from bivariate analyses with higher *p*-values of less than 0.25 were selected. This cut-off point of 0.25 is supported by the literature [98]. Hierarchical general linear model analyses were used to explore the significant predictors for caregiver burden, and depressive and anxiety symptoms in caregivers of PwD. Reference groups for all non-continuous independent variables are indicated in brackets in the tables.

#### 3.3.1. Multivariate Analysis on Predictors of Caregiver Burden in Caregivers of PwD

Table 10 illustrates the hierarchical general linear model analysis for predictors of caregiver burden in caregivers with PwD. The general linear model deduced that four variables significantly predicted caregiver burden after accounting for the effect of other variables, notably patient’s education level, caregiver’s employment status, types of dementia, and psychological flexibility. Firstly, lower education attainment in the PwD was a significant predictor of caregiver burden in caregivers of PwD (*p* = 0.04). In addition, in comparison with caregivers with full-time employment, unemployment was a significant predictor of caregiver burden (*p* = 0.00). Nevertheless, patients with MVAD and dementia of other aetiologies apart from Alzheimer’s disease and vascular disease, significantly predicted caregiver burden, compared with patients with Alzheimer’s disease (*p* = 0.01 and 0.04, respectively). Lastly, psychological inflexibility was a significant predictor of caregiver burden in caregivers of PwD (*p* = 0.00), where each unit increase in the score of psychological inflexibility rated by AAQ-II increased the ZBI score by 1.36 units. On the other hand, mindfulness did not significantly predict the caregiver burden in dementia caregivers.

#### 3.3.2. Multivariate Analysis on Predictors of Depressive Symptoms in Caregivers of PwD

Four variables were identified as significant predictors of depressive symptoms in dementia caregivers: patient’s ethnicity, caregiver’s age, caregiver’s gender, and psychological flexibility. Patients of other ethnicities (i.e., apart from the three major ethnic groups) significantly predicted depressive symptoms in caregivers of PwD (*p* = 0.02). Caregivers of younger age was also a significant predictor of depressive symptoms (*p* = 0.03). In addition, female gender predicted significant depressive symptoms among caregivers (*p* = 0.04). Lastly, psychological inflexibility was also a significant predictor of depressive symptoms in dementia caregivers, where each unit score increase in psychological inflexibility on AAQ-II raised the score of caregiver’s depressive symptoms in PHQ−9 by 0.32 units (*p* = 0.00). The general linear model analysis did not otherwise find a significant relationship between mindfulness and depressive symptoms in caregivers. The results of this analysis are displayed in Table 11.

#### 3.3.3. Multivariate Analysis on Predictors of Anxiety Symptoms in Caregivers of PwD

As displayed in Table 12, four variables which significantly predicted anxiety symptoms in caregivers of PwD were identified, namely caregiver’s age, caregiver’s education level, types of dementia, and psychological flexibility. Younger age and lower education level were significant predictors of anxiety symptoms in caregivers (*p* = 0.04 and 0.02, respectively). Vascular dementia was another significant predictor of anxiety symptoms compared with Alzheimer’s disease (*p* = 0.04). Lastly, once again, psychological flexibility significantly predicted anxiety symptoms in caregivers of PwD (*p* = 0.00) where every unit increment in AAQ-II increased the score in GAD-7 by 0.33 units. Mindfulness, on the other hand, failed to predict anxiety symptoms in dementia caregivers.

## 4. Discussion

The present study aimed to determine the relationship of psychological flexibility and mindfulness to caregiver burden, and depressive symptoms and anxiety symptoms in caregivers of PwD. The importance of the psychological aspects of dementia caregivers in explaining their adverse mental health outcomes have not been commonly explored previously. This study also aimed to determine the predictors of caregiver burden, and depressive and anxiety symptoms. The findings could provide information for further screening and early intervention for these outcomes.

First and foremost, the study aimed to identify the related factors and predictors of caregiver burden in caregivers of PwD. In bivariate analysis, none of the sociodemographic or illness characteristics was found to have a significant relationship with caregiver burden in caregivers of PwD. However, three of the factors were identified as significant predictors of caregiver burden, namely, the patient’s education level, caregiver’s employment status, and types of dementia. Firstly, lower education level was a significant predictor for caregiver burden in caregivers of PwD. This finding was consistent with other studies, where a higher education level among the PwD was associated with a lower caregiver burden [36,51]. This phenomenon could be explained by the psychopathology of brain plasticity, where its development over the years of education might either prevent the occurrence of behavioural disturbances or reduce the severity if they occur [51]. This is consistent with the cognitive reserve hypothesis, where it was hypothesized that the threshold for the expression of psychiatric symptoms associated with the neuropathology of the disease increases with more years of education, which also reduce the incidence of delirium [88]. A few studies have managed to identify a significant association between the presence of psychotic symptoms, notably hallucination and delusion, in PwD with fewer years of education, particularly in patients with mild or moderate Alzheimer’s disease [99,100]. This suggests that the protective effect of years of education on mental functioning might not be sufficient to counter the higher level of neurological dysfunction in severe dementia due to more extensive cortical damage, as evidenced in brain imaging. Furthermore, premorbid reservation of cognition and involvement of complex cognitively stimulating activities following a higher level of education attainment is beneficial in preventing or delaying the onset of dementia, particularly Alzheimer’s disease [101,102,103,104]. Slower disease progression is associated with better preservation of intellectual insight, which might reduce caregiver burden due to the PwD having less impaired awareness of their deficits [105].

Secondly, unemployment significantly predicted caregiver burden in PwD, in comparison with full-time employment. Previous literature shows mixed findings about the associations between the caregiver’s employment status and caregiver mental health outcomes [106]. However, there were many studies from different study locations with findings similar to the current study. In Turkey, Jamaica and Ghana, unemployment is a significant predictor for caregiver burden [107,108,109]. Even though employment and juggling work and caregiving could contribute to strain and burden, being employed outside the home might bring potential benefits to caregivers [110]. Having multiple roles (e.g., employer and informal caregiver) might act as a buffer for dissatisfaction in one of the roles. Caregivers can have social interactions at work, and satisfaction with social interactions is associated with fewer psychological symptoms, lesser role overload, and more gain from the caregiving experience [8,111]. Employment provides more opportunities for caregivers to experience ‘life outside caregiving’, which could be explained by the activity restriction model [112,113].

Another possible explanation is related to financial status and duration of caregiving. Unemployed caregivers might experience more financial constraints than employed caregivers, and the caregiving of PwD can lead to more financial burden [114]. Employed caregivers are less likely to be full-time primary caregivers, thus their caregiving time is shorter than unemployed caregivers. Previous studies found that caregiver burden increased with the duration of caregiving [115]. However, monthly household income, perceived financial status, and duration of caregiving were not found to have a significant relationship to or to be a significant predictor of caregiver burden in this study.

Different types of dementia also significantly predicted caregiver burden, namely, MVAD and dementia due to other aetiologies (i.e., apart from Alzheimer’s and vascular dementia) compared with Alzheimer’s disease. The finding that MVAD is a predictor of caregiver burden could be explained by the cognitive phenotypes of both aetiologies [116]. Studies have reported lower cognitive performance in patients with MVAD than in patients with Alzheimer’s disease, particularly in the cognitive domains of attention, memory, visuoconstruction, denomination, and executive functions [117,118,119]. More significant impairments in memory, language and verbal fluency, denomination, and constructive praxis were found in Alzheimer’s patients who suffer from cerebral infarction [119,120,121]. The complexity of dementia with multiple aetiologies, i.e., MVAD compared with a single aetiology such as Alzheimer’s dementia, contributes to more significant deterioration in various cognitive domains, thus worsening the caregiver burden.

Furthermore, dementia with other aetiologies (i.e., frontotemporal lobar degeneration, Parkinson’s disease, traumatic brain injury) significantly predicted caregiver burden as well. Previous literature has reported that frontotemporal dementia is significantly associated with caregiver burden and psychological morbidity compared with dementia of other aetiologies [122,123,124]. People with frontotemporal dementia are more likely to experience frequent and serious BPSD, e.g., agitation, disinhibition, apathy and eating abnormalities [125,126,127,128]. The level of caregiver burden in caregivers of Parkinson disease dementia was found to be higher than Alzheimer’s disease [129]. This is due to more serious functional deficits, particularly problems from severe motor and behavioural symptoms in patients with Parkinson’s Disease Dementia. This distinct nature of clinical manifestations, particularly the exhibition of prominent disturbing behavioural changes, difficulties with higher order cognitive functions, and motor disorders, could be explained by the subcortical involvement of Parkinson’s disease (i.e., basal ganglia and brain stem) [130,131,132]. Otherwise, there is a lack of literature exploring the relationship between dementia secondary to traumatic brain injury and caregiver burden. However, despite the possible explanation for the finding that these types of dementia are a predictor for caregiver burden, it is important to note that in this study, the number of patients with the above diagnoses was very low, i.e., one patient with frontotemporal lobar degeneration, one patient with dementia secondary to traumatic brain injury, and two patients with Parkinson disease dementia. Therefore, these three aetiologies were collapsed into one category for analysis and its significance should be interpreted carefully.

The second outcome of this study was in relation to the depressive symptoms in caregivers of PwD. Considering depressive symptoms in caregivers of PwD, the bivariate analysis revealed four significant sociodemographic factors, notably, the patient’s age, caregiver’s age, caregiver’s employment status, and caregiver–patient relationship. Among these four variables, only the caregiver’s age remained as a significant predictor of depressive symptoms in dementia caregivers. Other significant predictors of depressive symptoms in dementia caregivers were the patient’s ethnicity and caregiver’s gender.

At the bivariate level, the first factor related to depressive symptoms among the caregivers of PwD was the patient’s age, where caregivers of younger PwD experience a higher level of depressive symptoms. This finding was consistent with those of previous studies [30,54,133,134]. Malaysia has an aging population, where its transition from aging to and aged society in just 24 years has followed the trend in other countries with high-income economies, e.g., Japan. More than 7% of the population was aged 65 years and above in 2020, and this is estimated to increase to 14% in 2044. These changes have brought new challenges to society, especially in employment, income security, healthcare, and aged care. Therefore, it has been proposed to gradually increase the minimum retirement age to 65 years, and the elderly have been encouraged to lengthen their working lives and even continue working beyond retirement age [135]. However, patients with dementia at a younger age are more likely to face psychosocial problems, notably, relational difficulties, family conflicts, unemployment and financial issues, and adverse experiences with the diagnostic process [136]. Family members, particularly spouses of PwD, need to take greater responsibility, and these changes in behaviour and roles in a family might lead to depressive symptoms [124,137,138]. The relationship between younger patients’ age and greater caregiver’s depression might be related to behavioural symptoms, the severity of illness, or years of caregiving, but the findings from the literature are inconsistent [133,139].

Full-time employed caregivers were found to be suffering from higher levels of depressive symptoms than those who were retired. As mentioned earlier, there are more studies exploring the influence of employment and unemployment on caregiver burden, and mixed results were identified [106]. However, the finding in the current study is consistent with other literature, where employed informal caregivers were significantly more depressed than informal retired caregivers [140]. This finding of a higher burden in full-time employed caregivers than among unemployed caregivers was explained above by the buffer effect, but the finding of higher levels of depressive symptoms in full-time employed caregivers compared with retired caregivers could be explained by the dual role strain hypothesis, which posits that caregivers face stress overload and difficulties in handling the obligations of their paid jobs in the labour force on top of providing care for their ill family members. They have inadequate time, energy, and resources to take care of different aspects of life and fulfil their work and caregiving responsibilities at the same time, not to mention that they have insufficient time to take care of themselves, particularly their physical and emotional health; therefore, they are at higher risk of developing depression [141,142]. Family stress that spills over to work was found to be significantly affecting their working performance. They were more likely to be absent for partial workdays, make more mistakes at work, face trouble in remembering things, reduce work responsibilities, turn down promotions, and terminate jobs to cope with caregiving responsibilities [143,144,145]. These impacts on a caregiver’s work and career eventually lead to poorer health and more psychological distress in midlife [146].

On the other hand, retired caregivers might have better financial reserves via retirement schemes than full-time employed caregivers who are still working hard to maintain their financial status; thus, retired caregivers suffer from lesser depressive symptoms. According to the Mercer CFA Institute 2021 Global Pension Index, out of 43 countries, Malaysia’s retirement system is ranked third in Asia after Singapore and Hong Kong and 23rd globally. There was only a slight decrease in the country’s overall index value from 60.1 in 2020 to 59.6% in 2021, despite the COVID-19 pandemic, indicating that our retirement system was relatively adequate, sustainable, and integrated globally [147]. There are also several positive aspects of caring in retirement. Most of the literature focuses on the adverse effects of caregiving, but the few studies focusing on the positive aspects of caregiving report that caregivers derive satisfaction throughout the caregiving process. They were better at identifying more ‘uplifts’ than ‘hassles’ or more ‘rewards’ than ‘difficulties’ related to caregiving [148,149]. Especially retired spousal caregivers believe it is preferable they provide care rather than others; thus, they prefer to provide care rather than take the risk of the alternatives. The satisfaction and happiness in caregiving, in general, are derived from showing their love for the care recipient [150,151,152].

The caregiver–patient relationship was identified to have a significant association with depressive symptoms in caregivers of PwD, where non-spousal caregivers suffered from significantly greater depressive symptoms than spousal caregivers. This finding contrasts with previous studies which found that spousal relationships were associated with greater depressive symptoms, strain, and psychological morbidity [8,54,134,153]. This could be explained by the age of the caregivers in our study as more than 90% of the non-spousal caregivers were children, children-in-law, and grandchildren. These groups of caregivers shared a common factor, which was younger age than spousal caregivers in general. Our study also identified that younger caregivers reported higher depressive symptoms than older caregivers. Nevertheless, younger age of caregivers was a significant predictor of depressive symptoms in caregivers of PwD, consistent with the previous literature [154,155]. This may be because younger caregivers lack psychological preparation and experience in caregiving as most of them take up the role without much physical, psychological or financial preparation. They are more likely to have children and careers while providing caregiving to ill family members; thus, the level of depressive symptoms are heightened [155,156]. The role reversal in caregiving, becoming “a parent of the parent”, is emotionally taxing and causes significant distress among the younger caregivers.

The caregiver’s female gender also significantly predicted depressive symptoms in caregivers of PwD. This finding was consistent with previous literature [11,30,134,157,158]. One explanation for the gender difference in depressive symptoms in dementia caregivers might be related to stress-coping theory [159]. Women tend to use effective coping strategies (e.g., fantasy, denial, escape, avoidance) more frequently than men, while men have been found to use more effective coping styles (e.g., problem-solving, acceptance, distancing) than women [160]. However, this finding might not necessarily reflect the variations between genders in caregiving experience. Even though all items in PHQ-9 exhibited negligible gender-differential item functioning, some items such as ‘sadness’ might be in conflict with the societal ideals of masculinity [161,162]. Therefore, male caregivers might be reluctant to report these symptoms.

Another predictor of depressive symptoms in caregivers was other ethnicities (apart from the three major ethnic groups in Malaysia). Different ethnic groups have their own cultures; however, they do not differ from one another to a great extent as they originated in Eastern cultures with values and a set of social practices meant to maintain intergenerational integrity and solidarity [163]. For example, Malay practice the cultural values of balas jasa (i.e., repaying parents), and Chinese hold strong beliefs in filial piety. These values are spread widely spread among different ethnicities in Malaysia due to sharing and exchange of cultures with positive moral values [163,164,165]. Therefore, our finding might not reflect the real variation in ethnicity, especially when the ethnic group category of “others” was of extremely small size (i.e., 4 out of 82), compared with other ethnicities.

This study also aimed to identify the factors and predictors of anxiety symptoms in caregivers of PwD. In the bivariate analysis, several sociodemographics and illness characteristics were found to have significant associations with anxiety symptoms in caregivers of PwD, notably, the caregiver’s age, caregiver’s employment status, caregiver–patient relationship, and illness severity. However, out of these four variables, only caregiver’s age remained as the significant predictor of anxiety symptoms in dementia caregivers. The other two predictors identified from multivariate analysis were caregiver’s education level, and types of dementia.

Firstly, caregiver’s employment status had a significant association with anxiety symptoms in caregivers of PwD, where unemployed caregivers suffered from significantly greater anxiety symptoms than retired caregivers. This could be explained by the same theory that retired caregivers have better financial reserves through savings and pension or retirement schemes than unemployed caregivers, who might be struggling from significant financial constraints. Income inadequacy triggers many uncertainties about living and caregiving in the future, and the worries can trigger a cascade of anxiety symptoms, even though the relationship between monthly household income and perceived financial status with anxiety symptoms in caregivers were not found to be significant [58]. Additionally, retired caregivers are generally, but not exclusively, older than those who are unemployed, so their life experiences and physical, psychological, and financial preparation might be better than with unemployed caregivers [154,155].

Similar to the relationship to depressive symptoms, our study found a significant relationship between the caregiver–patient relationship and anxiety symptoms in dementia caregivers, where non-spousal caregivers experienced greater anxiety symptoms than spousal caregivers. Despite its contradiction with previous literature, this finding could be explained by the age factor among non-spousal caregivers (younger population) [166,167]. Similar to the previous findings, a younger caregiver age was not significantly related to anxiety symptoms, but it was one of the predictors of anxiety symptoms. There were similar findings in previous research [168,169]. The sudden addition of a caregiving role on top of the pre-existing responsibilities as a non-elderly adult comes with worries and uncertainties about the future due to inadequate physical and psychological preparation [155,156]. Therefore, the caregivers tend to feel anxious in the caregiving process.

Anxiety symptoms in non-spousal caregivers could be explained by another unique concept describing the relationship between female dementia patients and their caregivers, particularly daughters, namely “grateful guilt.” “Grateful guilt” is a concept developed by Ward-Griffin et al. (2006) to describe the conflicting and clashing emotional state of mothers with dementia [170]. They feel grateful for being helped by their daughters, but there is a strong desire to not be a burden to their families [170,171]. They want to retain and maintain a certain degree of control and independence over the relationship between caregiving and care-receiving and maintain their dignity as an adult [171,172,173]. Mothers may withhold information from their daughters regarding their feelings and make fewer demands despite necessity, while the daughter struggles to ensure the mother’s safety [173]. The tension and conflict between both parties may trigger anxiety symptoms among the caregivers.

Patient’s disease severity was another factor that had a significant relationship to anxiety symptoms in caregivers, where the milder the illness was, the greater the anxiety symptoms. The possible explanation for our finding is that at the early stage of dementia, the patients might often complain about their forgetfulness and declining memory, which is reflected by frequent questioning and impairment in activities of daily living and draws the attention of the caregivers as such complaints did not arise in the past [21]. The caregivers might also fear losing their self-identity as they start to feel the effects of aging due to memory deficits, although these are a normal part of the aging process. They may attempt to reduce the risk of dementia and modify their lifestyle [174]. The uncertainties of the illness progression, the fear of drastic illness deterioration, and the anticipation of poor prognosis trigger worries and fear among the caregivers. As the disease progresses, caregivers become more familiar with the disease and are able to accept the progression better; thus, in our study, the anxiety symptoms were lower when the patients reached the severe stage of dementia.

In terms of predictors of anxiety symptoms in caregivers of PwD, besides the caregiver’s age that was discussed earlier, lower caregiver education level was another significant predictor for anxiety symptoms. The findings from previous studies vary. Some did not find significant associations between caregiver’s education level and anxiety symptoms, while others identified significant associations [21,175,176]. Lower educational attainment might contribute to lower levels of reasoning and problem-solving skills that result in poorer access to information and less effective use of that information to provide better quality care for PwD [177]. Moreover, poorer understanding by caregivers with lower education levels of the psychopathology of the illness may trigger anxiety symptoms due to fear and uncertainties.

Lastly, it was found that compared with Alzheimer’s disease, vascular dementia significantly predicted anxiety symptoms in caregivers of PwD. The literature that studies the relationship between the types of dementia and anxiety is very scarce. However, the anxiety could be explained by the uncertainty and unexpected abrupt occurrence of cognitive decline following a cerebrovascular accident. It is a drastic change for the caregivers to accept as the patients usually suffer from physical impairment and motor disturbances, i.e., paralysis of the limbs, swallowing problems, bowel and urinary incontinence, etc. [178]. The additional cognitive impairment and high prevalence of behavioural and psychological symptoms of dementia could trigger a higher level of anxiety symptoms among caregivers due to a lack of expectations and anticipation.

The main objective of the study wa” to ’dentify the relationship Of psychological flexibility and mindfulness to caregiver burden, and depressive and anxiety symptoms in caregivers of PwD. Psychological inflexibility was found to be not only significantly related to but also a significant predictor of caregiver burden, and depressive and anxiety symptoms in caregiver of PwD, after controlling the confounders via multivariate analysis.

As mentioned earlier, psychological flexibility is the ability to act in accordance with long-term values and goals, despite inner discomfort often related to experiential avoidance and control [73]. Experiential avoidance, also known as emotional avoidance, is the process when a person is unwilling to remain in contact with unpleasant and unwanted internal or private experiences (e.g., bodily sensations, emotions, thoughts, memories, behavioural predispositions, etc.) and makes attempts to either alter or avoid the form or frequency of these experiences [179,180]. A person may avoid the aversive events or try to change via different behavioural processes. This may include, for example, cognitive entanglement (i.e., excessive involvement and identification with negative cognitions) or failure to engage with adaptive behaviour due to being hyper-focused on the adverse experience and heightening of the perceived need for control over emotions and cognitions, even though this is always unsuccessful [180,181,182,183]. On the other hand, acceptance, the adaptive alternative to experiential avoidance, is a mental attitude of openness and receptivity, nonjudgement, and equanimity towards internal and external experiences [184,185,186]. It is about “actively contacting psychological experience—directly, fully, and without needless defense—while behaving effectively” [179]. Hayes et al. (2004) conducted a multi-site study to measure experiential avoidance and acceptance and action using the Acceptance and Action Questionnaire (AAQ) among 2400 participants from various backgrounds (e.g., undergraduate students, patients in health maintenance organizations, members of self-help groups, British civil servants, etc.) who were experiencing a range of distress [187]. They found that higher scores on the AAQ, which indicate a higher level of avoidance and lower level of acceptance, were associated with higher levels of psychiatric symptoms, particularly depressive symptoms, anxiety symptoms, and trauma-related symptoms.

Our findings on the relationship between psychological flexibility and caregiver’s burden, and depressive, and anxiety symptoms are consistent with previous literature [78,79,80]. Dementia, in general, is a progressive, non-reversible neurocognitive disorder. Alzheimer’s disease is a progressive disorder with a steady decline over the years; vascular dementia, even though the disease progression follows a step-ladder pattern, is still irreversible. Caregivers often experience grief and shock, and they might have little prior knowledge of the illness and the psychopathology, especially at the early phase of illness. Later, with psychoeducation, they may realise that the use of psychotropic drugs can only delay the progression of the disease and control the behavioural or psychological symptoms without being able to halter or cure the illness. Finding solutions to work through these difficulties requires time, effort, energy, and familial involvement. As the disease progresses, the caregiving tasks increase as caregivers needed to assist patients in more situations or daily living activities. Avoidance behaviour and being hyper-focused on the negative consequences of caregiving for the PwD may only bring more negative impacts to the caregiver’s own mental health. Acceptance, on the other hand, might be a central part of coming to terms with these changes. Acceptance does not equate to suffering passively or giving up hope for recovery. On the contrary, it is an active attitude to willingly be part of the experience, while engaging in or pursuing personally meaningful behaviour, and making space for depression, anxiety, anger, and grief [188,189]. It provides an ameliorating effect to caregiver strain [190]. It can reduce intrusive thoughts in caregivers in relation to intrapsychic strain, as explained by the suffering-compassion model in caregiver depression and the fear of losing self-identity [134,174,191]. Therefore, psychological flexibility acts as a buffer to caregiver burden and depressive and anxiety symptoms.

Furthermore, mindfulness was found to have significant relationships to caregiver burden, and depressive, and anxiety symptoms in caregivers of PwD. However, in contrast to psychological flexibility, it was not found to predict any of these three outcomes significantly.

First and foremost, the concept of mindfulness needs to be understood. Mindfulness is defined as the tendency to purposely bring to attention the present experience with a mental stance of receptivity and acceptance without any judgment [74,75]. In the conscious state of mindfulness, rather than the experience itself, the self is identified as the observer of the experience [186,192]. Even though there are many conceptualizations of mindfulness being offered, including unidimensional and multidimensional approaches, there are two components that are commonly described across different conceptualizations, which are the use of attention to monitor present moment experiences and a mental attitude of acceptance towards momentary experiences [193,194]. Borkovec (2002) described the gist of mindfulness in a very whimsical but insightful and cogent way on page 79, writing that it “allows me to let go of the illusory future and past and focus on the non-illusory present [195]”. On the other hand, mindlessness is related to the inability to direct or redirect attention to the present moment, and thus it increases the mental ruminations about the past and the future [196].

There is limited literature studying mindfulness and its relationship to mental health. Most are interventional studies that explore the effectiveness of mindfulness-based therapy [81]. Mindfulness practices have been found to show improvements in mental health but not consistently in all studies [197,198]. There is moderate evidence for anxiety, depression, and pain improvement; a little on stress/distress improvement and mental health-related quality of life; and no evidence on positive mood and physical symptoms (e.g., appetite, sleep, weight) [199]. Instead, some studies report increased negative affect after mindfulness-based intervention [200]. In terms of the dementia caregiver population, the finding in our study was consistent with the previous literature, where being mindful did not reduce the level of caregiver burden [82]. One study suggested that mindfulness might benefit mental health directly by lowering the anxiety and depressive levels in a more straightforward way instead of through the toll that the caregiver burden takes on psychological functioning, but this was not shown in our study as mindlessness did not predict anxiety and depressive symptoms among the caregivers. This is understandable because as mindfulness increases, the emotional state awareness increases. Being aware of the present, including the issues and challenges in taking care of PwD, might not reduce the burden level or the depressive and anxiety symptoms. Instead, one intervention targeting mindfulness did not focus on the symptoms to but addressed the importance of value living [201]. Therefore, symptoms were just the secondary impact to the primary focus. This intervention did not attempt to directly change or stop unwanted thoughts or feelings but to encourage the new development of a compassionate relationship with those experiences. Therefore, being mindful, i.e., focusing on the present, does not modulate the burden, or depression and anxiety levels of the dementia caregivers, but the addition of psychological flexibility, i.e., experiential acceptance of the changes, improves the adverse psychological outcomes.

The findings from this study could be applied in the clinical setting by recommending psychotherapy that targets psychological flexibility among caregivers of PwD. At present, various interventions are provided to caregivers of PwD to improve their mental well-being, ranging from psychoeducation, psychotherapeutic interventions, caregiver training, spiritual and religious support, etc. [202]. The NICE guidelines (2018) recommend psychoeducation and skills training for caregivers of PwD [3]. However, no single effective method that can reduce distress or improve a caregiver’s well-being has been identified, and most only showed modest beneficial impact [203,204]. With the current trend towards third-wave psychotherapy, mindfulness-based intervention (MBI) (i.e., mindfulness-based stress reduction (MBSR), mindfulness-based cognitive therapy (MBCT)) are relatively popular, and several interventional studies have reported their significant moderating effect on caregivers’ mental health [81]. However, the insignificant predictive effect of mindfulness and the significant predictive effect of psychological flexibility suggested that intervention targeting experiential avoidance might be more efficacious in reducing caregiver burden, i.e., acceptance and commitment therapy (ACT), which is a relatively unexplored form of psychotherapy for caregivers.

There are few studies analysing the efficacy of ACT or its components (e.g., mindfulness) in the caregiver population. However, some small pilot studies found a positive effect of ACT on the mental health of dementia caregivers. A recent study explored the feasibility and acceptability of ACT for caregivers of PwD. It identified a significant improvement in caregiver burden and anxiety levels after the 6-week course Telephone ACT Intervention for Caregivers (TACTICs), indicating the effectiveness of ACT in improving caregivers’ mental health [205]. Another two studies reported significant reductions in depressive and anxiety symptoms, burden, and stress, and an increment in positive aspects of caregiving and caregiver’s quality of life after ten sessions of online ACT teaching (either self-guided or instructor-guided) [206,207]. An earlier meta-analysis also found a moderate effect on depressive symptoms and quality of life, small to moderate effect on stress, and a small effect on anxiety [207]. Compared with other established non-pharmacological interventions, very scarce literature is available. However, one study identified a similar efficacy of ACT and cognitive behavioural therapy (CBT) in improving depressive and anxiety symptoms among caregivers; therefore, ACT seems to be a viable and effective intervention for dementia caregivers, but more large-scale intervention studies are needed in the future [206].

There were a few limitations in this research that should be mentioned. First and foremost, this was a cross-sectional study, which did not allow us to investigate the causal relationship between the psychological factors and caregivers’ mental health, i.e., burden, depressive symptoms and anxiety symptoms. Other longitudinal study designs would be desirable to see the long-term effect of mindfulness and psychological flexibility on the level of burden, depression, and anxiety among dementia caregivers. Secondly, the sample size of the current study was small (*n* = 82) due to limitations resulting from the COVID-19 pandemic and the sampling method which eventually became universal sampling. Therefore, selection bias in the sample recruitment process might be present. Caregivers who were significantly burdened, depressed, or anxious might not be motivated to participate in this study, and those who were participating might be more mindful of their situation and well-being. Future studies with larger sample sizes should be conducted. Randomization should also be included to overcome selection bias.

There were also some confounding variables in this study that might influence both the dependent and independent variables. For example, employment status might influence both perceived financial status and caregiver burden. The effects of non-caregiving-related stress, e.g., work and marital issues on caregiver burden might be present as well. Therefore, besides performing multivariate analysis, future studies should attempt to decrease the impact of confounding variables by performing restriction, matching, or randomization, according to the study design. Lastly, our study only investigated the relationship of two psychological aspects of caregivers in PwD, notably psychological flexibility and mindfulness, to caregiver’s burden. Other psychological factors might play a significant role in modulating the outcome, such as coping skills, resilience, etc., which were not investigated in this study. In addition, other caregiver mental health outcomes were not identified either, e.g., depression, anxiety and quality of life. Therefore, it is recommended that in the future, other psychological factors and outcomes could be added to obtain a better and more comprehensive picture that reflects the whole dementia caregiving process.

## 5. Conclusions

Providing care for PwD is a challenging task and is related to various adverse mental health outcomes among caregivers, especially the caregiver burden. The predictive sociodemographics and illness characteristics identified in this study can assist clinical practitioners to screen these caregivers at risk. This study indicated that psychological inflexibility was a significant predictor of dementia caregiver burden when controlled. Therefore, by identifying their psychological flexibility, intervention programs that target the awareness of the caregiver’s psychological inflexibility should be implemented to reduce caregiver burden in dementia caregivers and improve their quality of life.

## Figures and Tables

**Table 1 ijerph-20-04232-t001:** Sociodemographics of PwD.

Variables	*n* (%)	Mean (SD)
Patient’s age (years)		77.84 (6.70)
Patient’s Gender		
Male	35 (42.7)	
Female	47 (57.3)	
Patient’s Ethnicity		
Malay	27 (32.9)	
Chinese	29 (35.4)	
Indian	22 (26.8)	
Others	4 (4.9)	
Patient’s Religion		
Islam	29 (35.4)	
Buddhism	22 (26.8)	
Christian	11 (13.4)	
Hinduism	16 (19.5)	
Others	1 (1.2)	
None	3 (3.7)	
Patient’s Education Level		
No formal education	21 (25.6)	
Primary education	23 (28.0)	
Secondary education	27 (32.9)	
Tertiary education	11 (13.4)	
Patient’s Marital Status		
Single	2 (2.4)	
Married	47 (57.3)	
Cohabitation	1 (1.2)	
Separated	3 (3.7)	
Divorced	2 (2.4)	
Widowed	27 (32.9)	

SD = Standard deviation.

**Table 2 ijerph-20-04232-t002:** Sociodemographics of caregivers of PwD.

Variables	*n* (%)	Mean (SD)/Median (IQR)
Caregiver’s Age (years)		51.50 (14.63) ^1^
Caregiver’s Gender		
Male	26 (31.7)	
Female	56 (68.3)	
Caregiver’s Ethnicity		
Malay	27 (32.9)	
Chinese	29 (35.4)	
Indian	22 (26.8)	
Others	4 (4.9)	
Caregiver’s Religion		
Islam	29 (35.4)	
Buddhism	19 (23.2)	
Christian	13 (15.9)	
Hinduism	16 (19.5)	
Others	1 (1.2)	
None	4 (4.9)	
Caregiver’s Education Level		
No formal education	2 (2.4)	
Primary education	6 (7.3)	
Secondary education	35 (42.7)	
Tertiary education	39 (47.6)	
Caregiver’s Marital Status		
Single	20 (24.4)	
Married	56 (68.3)	
Cohabitation	1 (1.2)	
Separated	0 (0.0)	
Divorced	2 (2.4)	
Widowed	3 (3.7)	
Caregiver’s Employment Status		
Full-time employment	34 (41.5)	
Part-time employment	9 (11.0)	
Unemployed	22 (26.8)	
Retired	17 (20.7)	
Caregiver’s Monthly Household Income (USD)		725.85 (752.74) ^2^
Caregiver’s Perceived Financial Status		
Problematic	26 (31.7)	
Non-problematic	56 (68.3)	
Caregiver-Patient Relationship		
Spouse/Partner	21 (25.6)	
Sibling	1 (1.2)	
Child	49 (59.8)	
Child-in-law	5 (6.1)	
Grandchild	2 (2.4)	
Others	4 (4.9)	
Caregiving Duration (hours/day)		14.65 (8.39) ^1^
Sharing of Caregiving		
Shared	59 (72.0)	
Unshared	23 (28.0)	

SD = Standard deviation; IQR = Interquartile range. ^1^ = Mean (SD); ^2^ = Median (IQR).

**Table 3 ijerph-20-04232-t003:** Illness Characteristics of PwD.

Variables	*n* (%)	Mean (SD)
Illness Duration (years)		5.88 (3.64)
History of Psychiatric Ward Admission		
Never admitted before	76 (92.7)	
Admitted once or more	6 (7.3)	
Types of Dementia		
Alzheimer’s Disease	39 (47.6)	
Vascular Dementia	14 (17.1)	
MVAD	25 (30.5)	
Frontotemporal Lobar Degeneration	1 (1.2)	
Parkinson’s Disease	2 (2.4)	
Traumatic Brain Injury	1 (1.2)	
Disease Severity		
Mild	31 (37.8)	
Moderate	25 (30.5)	
Severe	26 (31.7)	
BPSD		
Absence	24 (29.3)	
Presence	58 (70.7)	

SD = Standard deviation.

**Table 4 ijerph-20-04232-t004:** Psychological Flexibility and Mindfulness of Caregivers of PwD.

Variables	*n* (%)	Mean (SD)
AAQ-II		22.45 (10.69)
MAAS		68.23 (16.62)

SD = Standard deviation.

**Table 5 ijerph-20-04232-t005:** Caregiver Burden, Depressive Symptoms and Anxiety Symptoms in Caregivers of PwD.

Variables	*n* (%)	Mean (SD)
ZBI		33.17 (18.74)
No significant burden	27 (32.9)	
Significant burden	55 (67.1)	
PHQ-9		7.10 (6.24)
No significant depressive symptoms	57 (69.5)	
Significant depressive symptoms	25 (30.5)	
GAD-7		5.41 (5.54)
No significant anxiety symptoms	65 (79.3)	
Significant anxiety symptoms	17 (20.7)	

SD = Standard deviation.

**Table 6 ijerph-20-04232-t006:** Relationships Between Patient Sociodemographics and Caregiver Burden, and Depressive and Anxiety Symptoms.in Caregivers of PwD.

Variables	Caregiver Burden			Depressive Symptoms			Anxiety Symptoms		
	Mean (SD)	ST	*p*	Mean (SD)	ST	*p*	Mean (SD)	ST	*p*
Patient’s age (years)		−0.07 ^1^	0.56		−0.22 ^1^	0.04 *		−0.22 ^1^	0.11
Patient’s Gender		−1.60 ^2^	0.11		−1.72 ^2^	0.09		0.60 ^2^	0.10
Male	29.37 (17.81)			5.74 (5.73)			4.23 (5.42)		
Female	36.00 (19.09)			8.11 (6.47)			6.30 (5.52)		
Patient’s Ethnicity		0.68 ^3^	0.58		1.86 ^3^	0.19		0.44 ^3^	0.73
Malay	35.15 (15.78)			8.93 (6.11)			6.07 (5.80)		
Chinese	35.34 (17.18)			5.34 (5.42)			5.69 (5.69)		
Indian	27.55 (21.87)			6.41 (5.99)			4.32 (5.04)		
Others	35.00 (30.01)			11.25 (10.81)			5.00 (6.63)		
Patient’s Religion		0.78 ^3^	0.57		1.80 ^3^	0.12		0.90 ^3^	0.49
Islam	36.76 (16.69)			9.72 (6.60)			6.34 (5.78)		
Buddhism	32.68 (20.19)			5.55 (6.17)			5.86 (6.37)		
Christian	33.45 (18.65)			4.648 (5.00)			2.73 (3.17)		
Hinduism	26.56 (21.93)			6.56 (6.20)			5.25 (5.47)		
Others	21.00 (-)			4.00 (-)			-		
None	40.33 (1.16)			4.67 (2.31)			5.67 (2.31)		
Patient’s Education Level		−0.20 ^1^	0.07		−0.11 ^1^	0.32		−0.22 ^1^	0.47
Patient’s Marital Status		1.00 ^2^	0.32		0.52 ^2^	0.61		0.44 ^2^	0.66
Single/separated/	35.62 (20.62)			7.64 (7.46)			5.74 (5.72)		
divorced/widowed									
Married/cohabitation	31.44 (17.29)			6.86 (5.69)			5.19 (5.46)		

SD = Standard deviation; ST = Statistical test; *p* = *p*-value. ^1^ = Pearson correlation; ^2^ = Independent *t*-test; ^3^ = One-way ANOVA. * = *p*-value < 0.05.

**Table 7 ijerph-20-04232-t007:** Relationships between Caregiver Sociodemographics and Caregiver Burden, and Depressive and Anxiety Symptoms in Caregivers of PwD.

Variables	Caregiver Burden			Depressive Symptoms			Anxiety Symptoms		
	Mean (SD)	ST	*p*	Mean (SD)	ST	*p*	Mean (SD)	ST	*p*
Caregiver’s age (years)	-	−0.15 ^1^	0.19	-	−0.36 ^1^	0.00 **	-	−0.27 ^1^	0.02 *
Caregiver’s Gender		−0.43 ^2^	0.67		−1.84 ^2^	0.07		−0.46 ^2^	0.65
Male	31.85 (17.26)			5.38 (5.38)			5.00 (5.06)		
Female	33.79 (19.50)			7.89 (6.49)			5.61 (5.79)		
Caregiver’s Ethnicity		0.68 ^3^	0.58		1.86 ^3^	0.19		0.44 ^3^	0.73
Malay	35.15 (15.78)			8.93 (6.11)			6.07 (5.80)		
Chinese	35.34 (17.18)			5.34 (5.42)			5.69 (5.69)		
Indian	27.55 (21.87)			6.41 (5.99)			4.32 (5.04)		
Others	35.00 (30.01)			11.25 (10.81)			5.00 (6.63)		
Caregiver’s Religion		0.99 ^3^	0.43		1.89 ^3^	0.11		1.14 ^3^	0.35
Islam	36.76 (16.69)			9.72 (6.60)			6.34 (5.78)		
Buddhism	33.74 (21.24)			5.58 (6.14)			5.58 (6.43)		
Christian	30.31 (16.91)			4.46 (4.52)			2.85 (3.13)		
Hinduism	26.56 (21.93)			6.56 (6.20)			5.25 (5.47)		
Others	21.00 (-)			4.00 (-)			-		
None	43.25 (6.65)			6.75 (5.38)			8.25 (4.72)		
Caregiver’s Education Level		−0.02 ^1^	0.87		−0.01 ^1^	0.97		−0.13 ^1^	0.25
Caregiver’s Marital Status		0.66 ^2^	0.51	7.64 (7.46) 6.86 (5.69)	0.52 ^2^	0.61	6.36 (5.40) 5.00 (5.60)	1.02 ^2^	0.31
Single/separated/	35.24 (19.02)								
divorced/widowed	32.26 (18.71)								
Married/cohabitation									
Caregiver’s Employment Status		1.67 ^3^	0.18		5.17 ^3^	0.01 *		3.97 ^3^	0.02 *
Full-time employment	31.21 (20.69)			10.00 (7.23)			6.12 (6.70)		
Part-time employment	39.67 (17.18)			7.23 (5.36)			6.33 (4.12)		
Unemployed	38.27 (17.61)			3.65 (3.02)			6.18 (4.98)		
Retired	27.06 (18.74)			7.10 (6.24)			2.53 (3.26)		
Caregiver’s Monthly Household Income	-	0.06 ^4^	0.58	-	0.08 ^4^	0.47	-	−0.04 ^4^	0.72
Caregiver’s Perceived Financial Status		0.30 ^2^	0.77		−0.17 ^2^	0.86		−1.02 ^2^	0.31
Problematic	34.08 (18.21)			6.92 (5.97)			4.50 (4.66)		
Non-problematic	32.75 (19.12)			7.18 (6.41)			5.84 (5.90)		
Caregiver–Patient		−1.69 ^2^	0.10		−2.99 ^2^	0.00 **		−2.08 ^2^	0.04 *
Relationship									
Spousal	27.29 (16.64)			4.29 (4.34)			3.29 (4.78)		
Non-spousal	35.20 (19.11)			8.07 (6.53)			6.15 (5.63)		
Caregiving Duration (hrs/d)	-	−0.10 ^1^	0.40	-	−0.17 ^1^	0.13		-	0.21
Sharing of Caregiving		0.39 ^2^	0.70		1.07 ^2^	0.29		−0.47 ^2^	0.64
Shared	33.68 (17.93)			7.56 (6.62)			5.59 (5.37)		
Unshared	31.87 (21.03)			5.91 (5.09)			4.96 (6.06)		

SD = Standard deviation; ST = Statistical test; *p* = *p*-value. ^1^ = Pearson correlation; ^2^ = Independent *t*-test; ^3^ = One-way ANOVA; ^4^ = Spearman correlation. * = *p*-value < 0.05; ** = *p*-value < 0.01.

**Table 8 ijerph-20-04232-t008:** Relationships between Patient Illness Characteristics and Caregiver Burden, and Depressive and Anxiety Symptoms in Caregivers of PwD.

Variables	Caregiver Burden			Depressive Symptoms			Anxiety Symptoms		
	Mean (SD)	ST	*p*	Mean (SD)	ST	*p*	Mean (SD)	ST	*p*
Illness Duration (years)	-	−0.06 ^1^	0.59	-	−0.07 ^1^	0.55	-	−0.13 ^1^	0.26
History of Psychiatric Ward Admission		−0.93 ^2^	0.36		−1.19 ^2^	0.24		0.04 ^2^	0.97
Never admitted before	32.63 (17.92)			6.87 (6.31)			5.42 (5.38)		
Admitted once or more	40.00 (28.48)			10.00 (4.82)			5.33 (8.02)		
Types of Dementia		1.65 ^3^	0.22		2.60 ^3^	0.06		2.01 ^3^	0.12
Alzheimer’s Disease	30.38 (19.52)			5.69 (6.14)			4.64 (5.03)		
Vascular Dementia	33.57 (19.13)			9.14 (6.13)			7.57 (6.50)		
MVAD	38.60 (17.73)			8.84 (6.24)			6.12 (5.72)		
Others	25.00 (10.74)			2.75 (2.22)			1.00 (2.00)		
Disease Severity		−0.17 ^1^	0.12		−0.13 ^1^	0.26		−0.25 ^1^	0.02 *
BPSD		0.78 ^2^	0.44		1.49 ^2^	0.14		−0.89 ^2^	0.93
Presence	34.21 (20.02)			7.66 (6.79)			5.38 (5.74)		
Absence	30.67 (15.28)			5.75 (4.51)			5.50 (5.15)		

SD = Standard deviation; ST = Statistical test; *p* = *p*-value. ^1^ = Pearson correlation; ^2^ = Independent *t*-test; ^3^ = One-way ANOVA. * = *p*-value < 0.05.

**Table 9 ijerph-20-04232-t009:** Relationships between Psychological Flexibility and Mindfulness with Caregiver Burden, and Depressive and Anxiety Symptoms in Caregivers of PwD.

Variables	Caregiver Burden		Depressive Symptoms		Anxiety Symptoms	
	ST	*p*	ST	*p*	ST	*p*
AAQ-II	0.74 ^1^	0.00 **	0.61 ^1^	0.00 **	0.69 ^1^	0.00 **
MAAS	−0.37 ^1^	0.00 **	−0.43 ^1^	0.00 **	−0.43 ^1^	0.00 **

ST = Statistical test; *p* = *p*-value. ^1^ = Pearson correlation. ** = *p*-value < 0.01.

**Table 10 ijerph-20-04232-t010:** Hierarchical General Linear Model Analysis for Predictors of Caregiver Burden in Caregivers of PwD.

Variables	R^2^	Adj. R^2^	β (95% CI)	*p*	Tolerance	VIF
Step 4	0.69	0.63				
Patient’s gender (Female)			−3.45 (−9.84, 0.95)	0.29	0.69	1.46
Patient’s education level			−3.20 (−6.21, −0.19) *	0.04	0.76	1.32
Caregiver’s age			−0.17 (−0.45, 0.12)	0.24	0.44	2.26
Caregiver’s employment status (Unemployment)					0.81	1.24
Full-time employment			−10.42 (−17.42, −3.41) **	0.00		
Part-time employment			−4.79 (−14.72, 5.14)	0.34		
Retirement			−8.19 (−16.80, 0.42)	0.06		
Caregiver–patient relationship (Spousal)			−3.66 (−14.01, 6.69)	0.48	0.37	2.67
Types of dementia (Alzheimer’s disease)					0.89	1.13
Vascular dementia			5.92 (−1.84, 13.68)	0.13		
MVAD			7.95 (1.75, 14.15) *	0.01		
Other aetiologies			13.83 (0.35, 27.32) *	0.04		
Disease severity			1.86 (−1.59, 5.31)	0.29	0.85	1.18
Psychological flexibility			1.36 (1.07, 1.65) **	0.00	0.71	1.41
Mindfulness			−0.02 (−0.19, 0.15)	0.82	0.81	1.23

R^2^ = Percentage of variance explained by the model; Adj R^2^ = Adjusted percentage of variance explained by the model; β = Standardized regression coefficient; CI = Confidence interval; *p* = *p*-value; VIF = Variance inflation factor. Note: VIF > 5 suggests a collinearity issue. * = *p* < 0.05; ** = *p* < 0.01.

**Table 11 ijerph-20-04232-t011:** Hierarchical General Linear Model Analysis for Predictors of Depressive Symptoms in Caregivers of PwD.

Variables	R^2^	Adj. R^2^	β (95% CI)	*p*	Tolerance	VIF
Step 4	0.71	0.58				
Patient’s age			−0.04 (−0.24, 0.16)	0.68	0.58	1.74
Patient’s gender (Female)			−1.30 (−3.76, 1.15)	0.29	0.68	1.47
Patient’s ethnicity (Malay)					0.46	2.02
Chinese			6.22 (−5.09, 17.53)	0.28		
Indian			6.94 (−5.35, 19.23)	0.26		
Others			7.59 (1.09, 14.10) *	0.02		
Caregiver’s age			−0.15 (−0.29, −0.02) *	0.03	0.30	3.29
Caregiver’s gender (Female)			−2.48 (−4.88, −0.07) *	0.04	0.78	1.27
Caregiver’s religion (Islam)					0.52	1.94
Buddhism			−7.63 (−18.59, 3.33)	0.17		
Christian			−6.62 (−18.39, 5.14)	0.26		
Hinduism			−6.75 (−19.20, 5.70)	0.28		
Others			−1.32 (−14.65, 12.01)	0.84		
None			−8.96 (−20.81, 2.89)	0.14		
Caregiver’s employment status (Unemployment)					0.68	1.46
Full-time employment			−0.42 (−3.19, 2.36)	0.77		
Part-time employment			0.73 (−3.08, 4.55)	0.70		
Retirement			−0.56 (−4.29, 3.17)	0.76		
Caregiver–patient relationship (Spousal)			−1.30 (−6.13, 3.53)	0.59	0.24	4.16
Caregiving duration			0.04 (−0.13, 0.22)	0.62	0.50	2.01
History of psychiatric ward admission (Presence)			0.31 (−3.47, 4.09)	0.14	0.88	1.14
Types of dementia (Alzheimer’s disease)					0.86	1.17
Vascular dementia			2.18 (−0.71, 5.08)	0.14		
MVAD			1.99 (−0.35, 4.32)	0.09		
Other aetiologies			0.30 (−5.59, 6.19)	0.92		
BPSD (Presence)			1.40 (−0.83, 3.62)	0.21	0.87	1.15
Psychological flexibility			0.32 (0.22, 0.43) **	0.00	0.76	1.31
Mindfulness			= 0.06 (−0.13, 0.01)	0.08	0.72	1.38

R^2^ = Percentage of variance explained by the model; Adj R^2^ = Adjusted percentage of variance explained by the model; β = Standardized regression coefficient; CI = Confidence interval; *p* = *p*-value; VIF = Variance inflation factor. Note: VIF > 5 suggests a collinearity issue. * = *p* < 0.05; ** = *p* < 0.01.

**Table 12 ijerph-20-04232-t012:** Hierarchical General Linear Model Analysis for Predictors of Anxiety Symptoms in Caregivers of PwD.

Variables	R^2^	Adj. R^2^	β (95% CI)	*p*	Tolerance	VIF
Step 4	0.65	0.57				
Patient’s age			0.01 (−0.14, 0.17)	0.87	0.64	1.56
Patient’s gender (Female)			−0.81 (−2.87, 1.25)	0.44	0.70	1.43
Caregiver’s age			−0.11 (−0.22, −0.01) *	0.04	0.29	3.51
Caregiver’s education level			−1.75 (−3.18, −0.33) *	0.02	0.63	1.59
Caregiver’s employment status (Unemployment)					0.69	1.45
Full-time employment			−0.43 (−2.75, 1.89)	0.71		
Part-time employment			−1.03 (−4.25, 2.19)	0.52		
Retirement			−1.72 (−4.74, 1.29)	0.26		
Caregiver-patient relationship (Spousal)			0.65 (−3.14, 4.45)	0.73	0.26	3.91
Caregiving duration			0.06 (−0.08, 0.19)	0.41	0.51	1.94
Types of dementia (Alzheimer’s disease)					0.88	1.14
Vascular dementia			2.67 (0.15, 5.18) *	0.04		
MVAD			1.09 (−0.89, 3.07)	0.27		
Other aetiologies			2.53 (−1.85, 6.91)	0.25		
Disease severity			−0.23 (−1.32, 0.87)	0.68	0.81	1.23
Psychological flexibility			0.33 (0.23, 0.42) **	0.00	0.68	1.48
Mindfulness			−0.05 (−0.11, 0.01)	0.06	0.80	1.25

R^2^ = Percentage of variance explained by the model; Adj R^2^ = Adjusted percentage of variance explained by the model; β = Standardized regression coefficient; CI = Confidence interval; *p* = *p*-value; VIF = Variance inflation factor. Note: VIF > 5 suggests collinearity issue. * = *p* < 0.05; ** = *p* < 0.01.

## Data Availability

The data presented in this study are available on request from the corresponding author. The data are not publicly available due to privacy and confidentiality.

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
