# Peer review of "Relationship of Psychological Flexibility and Mindfulness to Caregiver Burden, and Depressive and Anxiety Symptoms in Caregivers of People with Dementia"

_ijerph, 2023, doi:10.3390/ijerph20054232_

Round 1
Reviewer 1 Report
The manuscript is good.Abstract, Introduction, Methods and materials, Results , Discussion and Conclusion well presented.

Reviewer 2 Report
Review for article “Relationships of Psychological Flexibility and Mindfulness with Caregiver Burden, Depressive and Anxiety Symptoms in Caregivers of People with Dementia”
Authors: Khai Pin Tan et al.
Journal: International Journal of Environmental Research in Public Health
The purpose of the study described in the given manuscript is finding of relationships of psychological flexibility and mindfulness with caregiver burden, depressive and anxiety symptoms in caregivers of People with Dementia (PwD). The topic is important because population of elderly people is gradually increased therefore number of caregivers is growing as well. Author provided adequate literature review and justification of their research topic, tools, materials and methods. They provided detailed justification of all tools and statistical methods. The authors provided detailed results with relevant illustration (figures) that gave them opportunity to organize sound discussion and formulate conclusion. The authors provided information about weaknesses of their study, which cleared up almost all the questions I had.
Authors of the article provided relevant references. The article looks as robust work. .
Based on the foregoing, I recommend this article for publication as presented.
Author Response
Thank you, reviewer 2.
Reviewer 3 Report
Thank you for this important contribution to understanding and hopefully mitigating the challenge of care giver burden for PwD. The background literature clearly supports the research rationale, questions and methodology. You note that the small sample size (N =82) is a limitation. The instruments used for this study all have strong psychometric properties. I did not note evidence of a power analysis. Was a power analysis conducted to determine N for this study? If not, was there a rationale for that?
Informed, implied consent was received from the participants, who were the caregivers. How was ethics addressed for the provision of data about the PwD. Approximately 31 % had high disease severity and presumably would not have been able to consent to release of their data, as well as those with moderate disease. How was this addressed. Did all participants have legal rights to consent on behalf of PwD to share their data? What was the source of data for PwD? Was this data provided by the care giver or from a chart review of the PwD? Clarity on these items would ensure protection of vulnerable persons (PwD) and accuracy of PwD demographic data.
Reviewer 4 Report
This is a review of the manuscript titled: “Relationships of Psychological Flexibility and Mindfulness with Caregiver Burden, Depressive and Anxiety Symptoms in Caregivers of People with Dementia” which presents the results of a cross-sectional study conducted in Malaysia that examine psychological flexibility and mindfulness in caregivers. This is an important and relevant topic in psychological research.
The article is well organized. The authors have made extensive research on the recent and past relevant publications doing a good analysis of the literature, although, I believe that some paragraphs have an excessive number of references that can be understood with a smaller number of them, e.g., line 104.
The manuscript presents an integrated development of the sections, the authors answer the questions presented in the introduction. The methodology is clear and appropriate to test the hypothesis referred and the statistical analysis is well performed and presented. Also, the discussion is extremely detailed and answers questions that a reader generates in the process of reading the article, which gives a plus to the article.
The tables are sequentially ordered, starting with the demographic analysis, and relationships between group characteristics and finishing with the hierarchical general linear model analysis. Overall, the article is well written, and the results are convincing for readers that are interested in understanding how a psychological state can aid caregivers to help persons that present a disability, not only dementia.
Minor revisions:
Line 82. although you define ZBI in the abstract, please define it on this line.
Page 5, line 232. The information in this paragraph is included in the table, it would be good that in the paragraph only refer to years and gender and refer to the rest of the sociodemographic information in Table 1.
Page 6. line 246. Refer to Table 2 for the sociodemographic information of the caregiver, by doing this you can save space for the paragraph and you still will be very clear.
Line 255. I suggest converting the household income to dollars, to have a better understanding of the economic conditions of other countries.
Line 270. MVAD, please clarify what it stands for in the paragraph.
Line 457. In the introduction, it would be important to refer to a couple of concepts, educational attainment, and cognitive reserve.
Line 476. Check “However, there were many studies from different study locations found similar findings as current study”, please specify them.
